

# A retrospective study on safety and clinical outcomes of unilateral biportal endoscopic technique for spinal degenerative diseases

Haitao Sun[1,2], Qi Zhang[1], Kehan Xu[1], Zihuan Zhou[1], Dongjie Jiang[1], Shaohui He[1], Jilu Liu[2] and Haifeng Wei[1]

[1] Department of Spinal Tumor, Changzheng Hospital, Naval Medical University CN, Shanghai, China
[2] Department of Spinal Surgery, Naval Hospital of Eastern Theater, Zhoushan, China

Corresponding authors
Jilu Liu, liujilu0717@163.com
Haifeng Wei, weihfspine@126.com

## ABSTRACT

**Background and Objective**. Unilateral biportal endoscopy (UBE) has been developing rapidly in coincidence with the popularization of minimally invasive spine surgery (MISS). However, the clinical outcome and invasiveness of UBE-assisted spinal surgery (UBESS) are undefined. The aim of the present study was to summarize the clinical outcome and surgical invasiveness of UBE for the treatment of various spinal degenerative diseases in a single center to validate the safety and application value of UBESS.

**Methods**. Included in this study were 105 patients who received UBESS from November 2021 to June 2022 in our center. All patients were followed up postoperatively for at least 12 months. All basic information was recorded to depict the demographic and surgical variables. Clinical outcomes were assessed in terms of the operation time, complications, days of hospital stay, total blood loss, intraoperative blood loss, postoperative drainage volume, hidden blood loss, biochemical changes associated with surgical injury, comparison of the visual analogue scale (VAS) for back and leg pain, Japanese Orthopedic Association (JOA) scores for cervical diseases at preoperative and postoperative stage, as well as Oswetry Disability Index (ODI), and modified MacNab scores one year after treatment.

**Results**. Of the 105 included patients, 68 patients were with single-level lumbar degenerative diseases, 22 with two-level diseases, two with three-level diseases, 10 with single-level isthmic spondylolisthesis, and three with single-level cervical spondylotic radiculopathy. UBE was performed by using five surgical approaches. The operation time, days of hospital stay, blood loss, postoperative immobilization time and prognosis were all estimated in consideration of the surgical approaches and the number of operated segments. The mean operation time ranged from 80 min to 214 min for single-level spinal diseases, and 112 min to 330 min for two-level ones. Total blood loss was higher in multilevel spinal surgery and single-level UBE-assisted lumbar interbody fusion+discectomy (ULIFD). The postoperative immobilization duration was between 0.5 and 2 days for single level spinal diseases, 1 and 3 days for 2-level diseases, fewer than 2 days for three-level diseases, and 1.5–12 days for isthmic spondylolisthesis (IS). The VAS and ODI for lumbar diseases decreased significantly and the JOA scores for cervical diseases improved after operation. The satisfaction rate was 89.70%, 85.71%, 90.00%, 66.67%, and 90.00% for the five surgical approaches respectively.

**Conclusions**. UBESS has proved to be a safe, reliable and minimally invasive option for spinal degenerative diseases, with significant benefits of pain control, rapid functional recovery, short hospitalization, and early rehabilitation. However, postoperative hidden blood loss should be put under the careful management when performing UBESS.

# INTRODUCTION

Spinal degenerative disorders (SDD), including intervertebral disc herniation/protrusion, spinal canal stenosis, spondylolisthesis and spondylosis, are one of the leading causes of chronic pain and radiculopathy, and a frequent indication for spinal surgery. Given the aging population, a more sedentary lifestyle, overall improvement of public health, and the evolution from contagious diseases to chronic diseases, the incidence of SDD is on the increase as expected (*Li et al., 2019*). The ascending trend renders more SDD patients as surgical candidates and more spinal surgeries in the past decades (*Tetreault et al., 2015*). Fenestration, laminectomy and lumbar fusion are common conventional open surgical options for SDD. However, they are disadvantaged by the high risk of infection, great damage to paravertebral soft tissues and the possibility of spinal instability. With the rapid evolvement of endoscopic platforms and techniques, minimally invasive surgery has emerged as an alternative surgical approach for SDD. Although minimally invasive spinal surgery reduces the surgical exposure and lowers the damage, the narrow working space limits the visualization and surgical performance.

Recently, unilateral biportal endoscopic spinal surgery (UBESS) has attracted increasing attention, for the instrumental channel and visual channel are independent of each other. Thus, unilateral biportal endoscopy (UBE) has the advantages of unrestricted instrument use and a wider visual field (*Heo et al., 2017*). Compared with percutaneous uniportal endoscopy, the improvement of operational flexibility enables surgeons to perform more precise and safer operations under indirect visualization. Some previous clinical studies have reported the application of UBE in single-level lumbar disc herniation, lumbar spondylolisthesis and cervical spondylopathy, but there is limited knowledge about its surgical invasiveness (*Jiang et al., 2022*; *Park et al., 2017*; *Kang et al., 2021*). Generally, total blood loss (TBL) is regarded as an indicator for surgical invasiveness. But it is admitted that overt blood loss, including intraoperative blood loss (IBL) and drainage volume, is inadequate to estimate TBL (*Lin et al., 2022*). Residual blood in the dead space and extravasation, which are referred to as hidden blood loss (HBL), are reported to significantly increase TBL and inversely decrease postoperative hemoglobin (Hb) (*Kang et al., 2021*). However, HBL is often overlooked in clinical practice, and continuous saline irrigation makes the estimation of blood loss difficult. Therefore, serological markers, creatine kinase (CK) and C-reactive protein (CRP) are introduced as objective indicators to assess surgical invasiveness simultaneously.

To the best of our knowledge, few studies have comprehensively described the clinical efficacy and surgical invasiveness of UBESS for single-level SDD, and there remains one significant gap in multilevel SDD. In the present study, we performed a detailed analysis with respect to the comparison of perioperative parameters and prognoses to evaluate the clinical outcomes of this technique in a large cohort of patients who were subjected to follow-up observations.

## MATERIALS AND METHODS

### Patient population and study design

Patients with clinically and radiographically confirmed SDD in our center from November 2021 to April 2022 were retrospectively reviewed. Pain intensity and functional disability were assessed to evaluate the pathological degree. Preoperative MRI and CT data were obtained for all candidates to confirm the surgical indications and select surgical strategies. The inclusion criteria were preoperative back/leg pain intensity ≥5 on Visual Analogue Score (VAS), intervertebral disc herniation/protrusion, spinal canal stenosis, spondylolisthesis and spondylosis undergoing UBESS, request of minimally invasive therapy, and follow-up with sufficient clinical data. Patients were excluded from the studies for the following reasons: lost to follow-up, contraindications, refusal of surgery, incomplete clinical data, and conservative treatment. Finally, a total of 105 patients were included in this retrospective study.

This research was approved by the Naval Hospital of Eastern Theater ethics committee (number: DHL202105), and informed consent was obtained from all participating patients or their delegated guardians. We adopted the method of having participating patients or their delegated guardians fill out informed consent forms and collected all the consent forms.

### Surgical technique

After general anesthesia, the patient was placed in a prone position on the radiolucent table to ensure that the target pathological region was perpendicular to the ground under C-arm fluoroscopy. The initial target anatomical site was the junction of the spinal process and lamina, and the inferior margin of the upper lamina was then confirmed under fluoroscopy and marked with a horizontal line. Two skin incisions were made roughly on the medial margin of the upper and lower pedicles, 1−1.5 cm cranial or caudal to the marked line. The lower incision was for the working portal, and the upper one was for the endoscopic portal. The core dilators were inserted from the working and endoscopic portal through interfascicular space to the junction of the spinal process and lamina. Serial dilators and periosteal strippers were used to expand the corridor and space. Once the triangulation for observation and instrument channel was made, the endoscope combined with the irrigation system and radiofrequency was placed within the established corridor and space. The soft tissue on the lamina and facet was peeled off to expose the inferior margin of the upper lamina and articular process. Sustained hemostasis by radiofrequency was conducted during operation to maintain a clear view. Next, partial laminectomy was performed to remove part of the inferior lamina of the upper lamina until the upper free margin of the

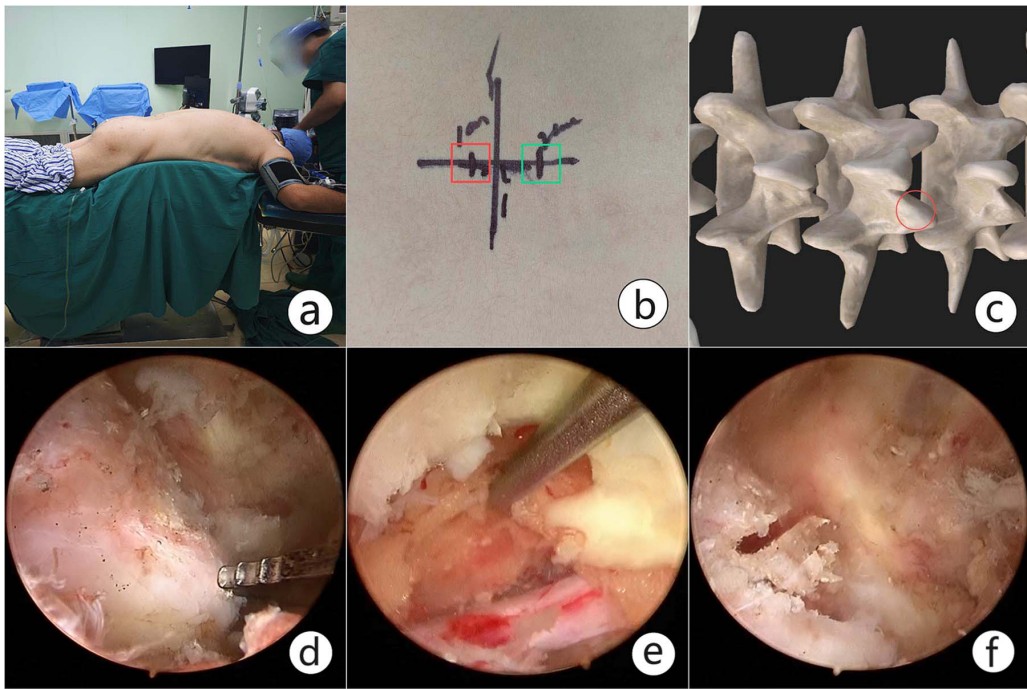

**Figure 1 An intraoperative view of unilateral biportal endoscopy spinal surgery (UBESS).** (A) The specific position for UBESS. (B). Skin marks for the left approach in lumbar surgery. The instrumental portal (red box) is cranial to the horizontal line, and endoscopic portal (green box) is caudal to the line. (C) The instrumental and endoscopic portals are confirmed over the junction of the spinal process and lamina. (D). Exposure of the inferior margin of the lamina and articular process. (E) Partial laminectomy was performed to dissociate the initial origin of the ligamentum favum. The dural sac and nerve root were confirmed after sufficient flavectomy. (F) Partial articular process resection. It is necessary to protect the nerve and decompress the transverse nerve root.

ligamentum favum was exposed. The lower margin of the ligamentum favum was detached from the lower lamina with pituitary forceps. Part of the articular process was resected with an electrical drill, Kerrison punch and osteotome, which is necessary for nerve protection (Fig. 1).

For patients with lateral recess stenosis, the lateral recess and nerve root canal were decompressed by ipsilateral flavectomy. For patients with bilateral stenosis, it is necessary to remove the lamina sufficiently to the contralateral side, and the ligamentum flavum to expose the contralateral facet joint and lateral recess for contralateral decompression (Fig. 2).

For patients with lumbar/cervical disc herniation, the ruptured disc was removed routinely, the soft disc particle was removed with pituitary forceps, and the calcified disc material was removed with a Kerrison punch. Internal disc decompression was recommended to reduce the risk of recurrence and chemical inflammation. After discectomy, annuloplasty was performed with a disposable fiber loop suture device (Jinxinxing, Inc., Beijing, China) for a certain group (Fig. 3).

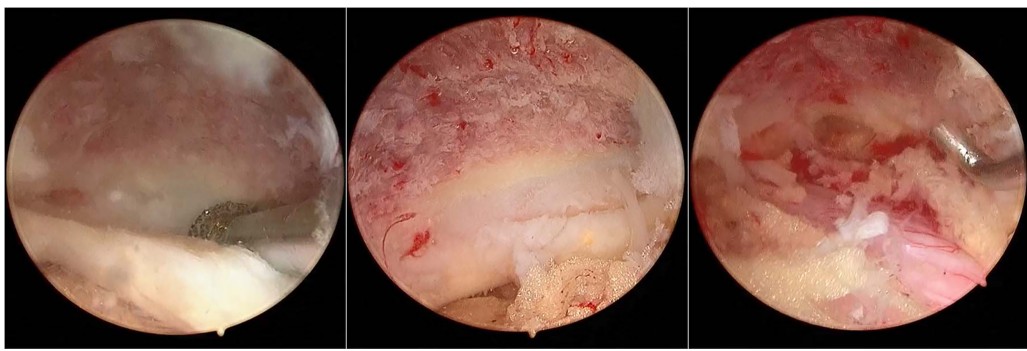

**Figure 2 Additional midline laminectomy was performed over the ligamentum favum midline.** It is necessary to remove the lamina sufficiently to the contralateral foramen and the ligamentum favum for the sake of enlarging the canal volume, thus achieving bilateral decompression.

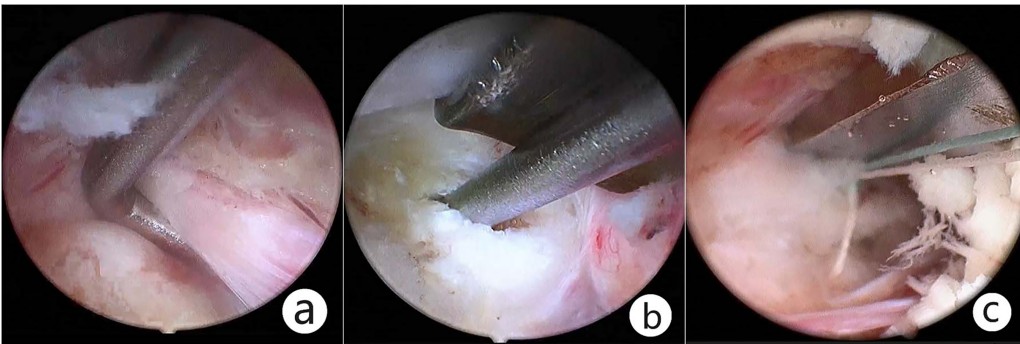

**Figure 3 Resect the ruptured disc and suture the disrupted fibrous annulus.** (A) Starting from the origin of the transverse root, dissection was proceeded between the nerve and the disc. (B) The ruptured disc was removed through the ipsilateral axillary corridor. Further internal decompression was accomplished with the use of pituitary forceps; (C) Annuloplasty, namely, disrupted fibrous annulus was sutured with a disposable fiber loop suture device.

For patients with lumbar spondylolisthesis, the intervertebral space was processed with a reamer and pituitary forceps after discectomy. Then the cartilage endplate was curetted for interbody bone grafting through an infundibular bone graft device. Autologous bone, or allografts were used to promote bony fusion. An interbody cage of appropriate size was placed under endoscopic visualization. After that, the percutaneous screws were implanted into bilateral pedicles under C-arm fluoroscopy. Finally, the pedicle screw position was confirmed and the drainage tube was indwelled before closure (Fig. 4).

## Data collection

The data were collected from the hospital admission records up to May 2023. The general information included gender, age, height, weight, body mass index (BMI), symptom duration, cardinal symptom, and follow-up duration. Perioperative data included diagnosis, operative approaches, the number of operative segments, operative level, operation time, postoperative immobilization time, drainage tube indwelling time,

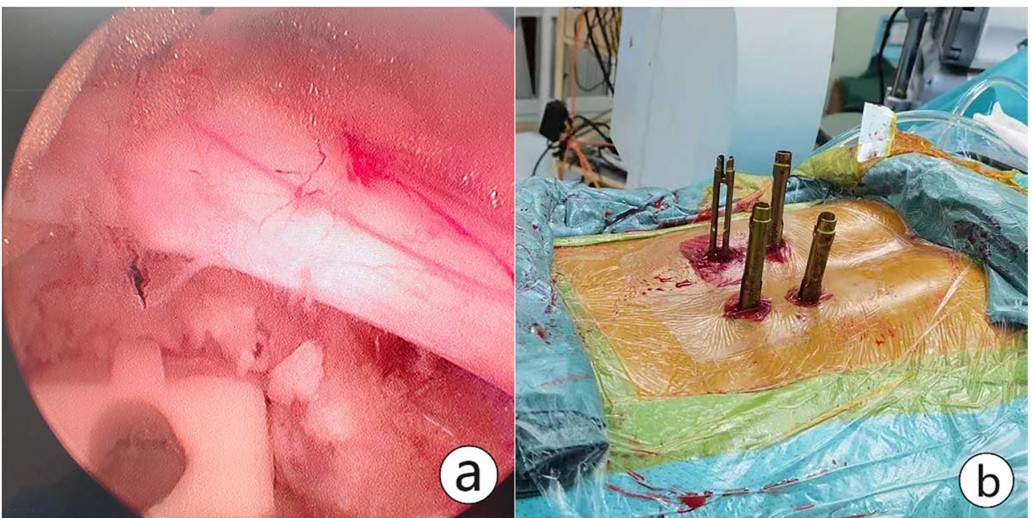

**Figure 4** **An interbody cage was placed.** (A) The cartilage endplate was processed for interbody bone grafting through an infundibular bone graft device. An interbody cage of appropriate size was placed under endoscopic visualization. (B) Percutaneous pedicle screws were implanted after cage insertion and confirmed. Postoperative CT showing UBE-assisted lumbar interbody fusion+discectomy.

postoperative hospital stay, complications, postoperative blood loss (PBL), TBL, IBL, HBL, hematocrit (Hct), Hb, CK and CRP.

Laboratory testing was regularly performed. Preoperative Hb, Hct, CK and CRP were recorded at admission. Postoperative Hb, Hct, CK and CRP were detected on the 2nd day after operation. No blood transfusion was required during or after operation, and thus TBL was calculated based on the perioperative change of Hct (*Nash et al., 2016*; *Gross, 1983*).

$$TBL(ml) = Patient\ blood\ volume(PBV, ml) \times \frac{(Hctpre - Hctpost)}{(Hctpre + Hctpost)} \times 2.$$

In this calculation, $Hct_{pre}$ and $Hct_{post}$ are short for pre- and postoperative Hct, respectively.

PBV (ml) = (k1 × height(m)$^3$ + k2 × weight(Kg) + k3) × 1000. (k1 = 0.3561, k2 = 0.03308, and k3 = 0.1833, for women; k1 = 0.3669, k2 = 0.03219, and k3 = 0.6041, for men) (*Nadler, Hidalgo & Bloch, 1962*).

IBL was determined by the sum of volume differences between suction fluid and irrigation liquid, and the amount of blood absorbed by gauze and towels using visual estimation (*Ali Algadiem et al., 2016*). PBL comprised the drainage flow and the blood loss visually estimated by soaked gauze. The drainage tube was removed when the drainage flow was less than 50 ml. HBL was calculated by Sehat's method (*Sehat, Evans & Newman, 2004*). Therefore, HBL=TBL-IBL-PBL.

Clinical outcomes, such as pain control, functional modification and patient satisfaction, were evaluated by VAS, Oswestry Disability Index (ODI), Japanese Orthopedic Association scores (JOA) and modified MacNab Score, respectively. Back and leg VAS (0–10), and ODI (0–100%) for lumbar disorders were investigated at admission, 1 month, 6 months and 1 year after operation. The improvement rate (%) of JOA (0–17) for cervical diseases

was calculated at the final follow-up [improvement rate (%) = (postoperative score –preoperative score)/(17- preoperative score)]. Plain radiography and CT were conducted 1 year after ULIFD. Interbody fusion was assessed *via* radiographical imaging at the final follow-up. Grade I and Grade II were identified as spinal fusion according to the Bridwell grading system (*Bridwell et al., 1995*). Patient satisfaction was rated according to the modified MacNab score at the final follow-up, with excellent and good defined as clinically satisfactory.

## RESULTS

### Basic demographics and perioperative information

Demographic information of the 105 patients who underwent UBESS in our center is shown in Table 1. They included 64 males (60.95%) and 41 females ranging in age from 20 to 79 years, with a mean of $50.57 \pm 17.04$ (median 51) years. 17.1% of the patients were younger than 30 years, and 12.4% of the patients were older than 70 years. The height and weight of the patients were collected routinely as basic admission information. The mean height was $1.69 \pm 0.08$ m, and the mean weight was $68.55 \pm 9.78$ kg. Forty-four patients (41.9%) with the complaints of pain and weakness were admitted as inpatients; 35 patients (33.33%) complained of pain and paresthesia and 26 patients (24.76%) with pain were outpatients of our hospital. The mean symptom duration was $4.03 \pm 5.66$ years. All the patients with progressive symptoms sought further treatment owing to the failure of conservative treatment.

The surgical information was obtained from the medical records. Of the 105 included patients, 77 (73.33%) were affected by lumbar disc herniation (LDH), 12 (11.43%) with spinal stenosis (SS), 10 (9.53%) with lumbar isthmic spondylolisthesis (LIS), three (2.86%) with cervical spondylotic radiculopathy (CSR), two (1.90%) with lumbar spinal stenosis (LSS), and one (0.95%) with recurrent LDH (RLDH) (Table 2). In consideration of the operative site and disease type, LDH, LDH with SS, LSS and RLDH were considered as the lumbar degenerative disorder (LDD) group. CSR and LIS were classified as an independent group. There were 68 patients with single-level LDD, 22 with two-level LDD, two with three-level LDD, 10 with single-level LIS, and three with single-level CSR. Five operative approaches were employed, including UBE-assisted lumbar discectomy+laminectomy+syndesmectomy (UDLS-L), UBE-assisted lumbar discectomy+laminectomy+syndesmectomy+annuloplasty (ULSA), UBE-assisted laminectomy+syndesmectomy (ULS), UBE-assisted cervical discectomy+laminectomy+syndesmectomy (UDLS-C), and UBE-assisted lumbar interbody fusion+discectomy (ULIFD), of which three (UDLS-L, ULSA and ULS) were applied for LDD, and two (UDLS-C and ULIFD) were used for CSR and LIS.

Compared with single-level LDD, the mean operation time (mOT) was shorter for multilevel LDD with UDLS-L or ULS, and longer for single-level LDD with ULSA than UDLS-L or ULS. All UBESSs were performed by an experienced spinal surgeon.

Regarding complications, pleural effusion occurred in one patient, small dural tear in one patient, and lateral femoral cutaneous nerve injury in one patient in LDD group, the

**Table 1  Demographics of patients receiving UBESS.**

| Demographics | UBESS |
|---|---|
| Gender, Male/Female (Male %) | 64/41 (60.95%) |
| Age (ys) | 50.57 ± 17.04 |
| Weight (kg) | 68.55 ± 9.78 |
| Height (m) | 1.69 ± 0.08 |
| BMI (kg/m$^2$) | 23.94 ± 2.23 |
| Cardinal symptom | |
|    Pain only | 26 (24.76%) |
|    Pain and paresthesia | 35 (33.33%) |
|    Pain and weakness | 44 (41.9%) |
| Symptom duration (years) | 4.03 ± 5.66 |

**Table 2  Perioperative diagnosis of patients receiving UBESS.**

| Perioperative diagnosis | UBESS |
|---|---|
| LDH | 77 (73.33%) |
| LDH+SS | 12 (11.43%) |
| LIS | 10 (9.53%) |
| CSR | 3 (2.86%) |
| LSS | 2 (1.90%) |
| RLDH | 1 (0.95%) |

complication rate being 3/92. All the patients with complications in this group received non-surgical treatment for the symptoms, and close follow-up was warranted. In the LIS group, cerebrospinal fluid leakage, wound infection, nerve root compression and pedicle penetration by screws were found in three patients, the complication rate being 3/10. Potent antibiotic, 3rd generation cephalosporin, was given for incision infection after bacterial culture and antimicrobial susceptibility tests. Revision surgery by UBE was performed to adjust pedicle screws and decompression. The complications occurred in the early period when UBESS was firstly carried out. All patients were discharged from our center after recovery. The detailed information is listed in Tables 3, 4 and 5.

## Comparisons of perioperative serology and blood loss

Overall, serological changes and blood loss in multilevel LDD were significantly higher than those in single-level LDD, and the serological parameters and blood loss were increased in the treatment of LIS with ULIFD as compared with UDLS-L, ULS, UDLSA and UDLS-C (Tables 6 and 7). There were approximate mean CK and CRP level changes in CSR and single-level LDD except ULS. The perioperative Hb and Hct differences were slightly obvious in LDD with UDLSA, but similar change was found between LDD with UDLS-L and with ULS. The mean TBL in ULS was 273.98 ± 42.59 ml, suggesting that it may be least invasive surgical approach for single-level LDD. The discrepancy was also observed in multilevel LDD. HBL was the major component to be reckoned with, which accounted

**Table 3  Surgical information of patients receiving UBESS.**

| | UBESS |
|---|---|
| | LDD (LDH/LDH+SS/LSS/RLDH) |
| Operative approach | |
| UDLS-L | 68 |
| UDLSA | 10 |
| ULS | 14 |
| Operative segment | |
| single-level LDD | 68 |
| two-level LDD | 22 |
| three-level LDD | 2 |
| Operative level | |
| L1/2 | 1 |
| L2/3 | 1 |
| L3/4 | 2 |
| L4/5 | 38 |
| L5/S1 | 26 |
| L1/2, L5/S1 | 1 |
| L2/3, L4/5 | 1 |
| L3/4, L4/5 | 10 |
| L4/5, L5/S1 | 10 |
| L3/4, L4/5, L5/S1 | 2 |
| Operation time (mins) | |
| UDLS-L (single/double/triple segments) | $110.88 \pm 13.75/150.31 \pm 18.50/254.50$ |
| UDLSA (single segment) | $129.40 \pm 7.35$ |
| ULS (single/double/triple segments) | $100.71 \pm 4.64/130.83 \pm 8.34/205.00$ |
| Postoperative immobilization time(days) | |
| UDLS-L (single/double/triple segments) | $1.36 \pm 0.76/1.44 \pm 0.31/2.00$ |
| UDLSA (single segment) | $1.35 \pm 0.34$ |
| ULS (single/double/triple segments) | $1.36 \pm 0.48/1.42 \pm 0.38/1.50$ |
| Drainage tube indwelling duration(days) | |
| UDLS-L (single/double/triple segments) | $0.78 \pm 0.30/0.84 \pm 0.30/1.00$ |
| UDLSA (single segment) | $0.80 \pm 0.26$ |
| ULS (single/double/triple segments) | $0.78 \pm 0.26/0.83 \pm 0.41/1.00$ |
| Postoperative hospital stay (days) | |
| UDLS-L (single/double/triple segments) | $3.49 \pm 1.19/3.84 \pm 0.77/5.00$ |
| UDLSA (single segment) | $3.45 \pm 0.93$ |
| ULS (single/double/triple segments) | $3.43 \pm 0.53/3.83 \pm 0.75/5.00$ |
| Complications | |
| Pleural effusion | 1 |
| Cerebrospinal fluid leakage | 1 |
| Lateral femoral cutaneous nerve injury | 1 |

for more than 70% of TBL in each group. The results of perioperative serology and blood loss are listed in Tables 6 and 7, respectively.

**Table 4   Surgical information of patients receiving UBESS.**

| Surgical information | UBESS |
|---|---|
| LIS | |
| Operative approach | |
| ULIFD | 10 |
| Operative segment | |
| 1 | 10 |
| Operative level | |
| L4/5 | 3 |
| L5/S1 | 7 |
| Operation time (mins) | 193.80 ± 14.13 |
| Postoperative immobilization time (days) | 1.80 ± 0.78 |
| Drainage tube indwelling duration (days) | 1.05 ± 0.32 |
| Postoperative hospital stay (days) | 5.30 ± 1.05 |
| Complications | |
| 1. Cerebrospinal fluid leakage | |
| 2. Pedicle penetration by screws | 1 |
| Nerve root compression | 1 |
| Wound infection | 1 |
| Postoperative hospital stay (days) | 3.42 ± 0.38 |

**Table 5   Surgical information of patients receiving UBESS.**

| Surgical information | UBESS |
|---|---|
| CSR | |
| Operative approach | |
| UDLS-C | 3 |
| Operative segment | |
| 1 | 3 |
| Operative level | |
| C4/5 | 1 |
| C5/6 | 2 |
| Operation time (mins) | 136.33 ± 16.62 |
| Postoperative immobilization time (days) | 1.33 ± 0.29 |
| Drainage tube indwelling duration (days) | 0.83 ± 0.29 |
| Postoperative hospital stay (days) | 3.42 ± 0.38 |

## Clinical outcomes of UBESS

The clinical outcomes and patient satisfaction are shown in Table 8. The mean follow-up duration of the series available was 13.14 ± 0.90 months. Postoperative back and leg VAS, ODI and JOA were significantly improved in each disease group. Improvements in pain intensity and lumbar dysfunction one month after operation were remarkable. The improvement rate was 100%, 100% and 66.67% for the three CSR patients at the final follow-up. Evaluation of interbody fusion using the Bridwell grading system showed a fusion rate of 100%. At the last follow-up, the clinical satisfaction rate according to

Table 6 **Changes of perioperative laboratory indices.**

| Variable | Operative segment | | |
|---|---|---|---|
| | Single segment | Double segment | Triple segment |
| Hct loss (%) | | | |
| UDLS-L | 2.31 ± 0.66 | 4.99 ± 1.34 | 7.00 |
| ULS | 2.71 ± 0.54 | 3.90 ± 0.51 | 3.20 |
| UDLSA | 3.12 ± 0.45 | – | – |
| UDLS-C | 2.10 ± 0.34 | – | – |
| ULIFD | 5.81 ± 1.28 | – | – |
| Hb loss (g/L) | | | |
| UDLS-L | 11.83 ± 8.83 | 17.25 ± 7.78 | 21.00 |
| ULS | 11.86 ± 5.84 | 14.50 ± 6.72 | 17.00 |
| UDLSA | 12.86 ± 2.64 | – | – |
| UDLS-C | 9.33 ± 1.53 | – | – |
| ULIFD | 20.90 ± 5.24 | – | – |
| CK change (1d, U/L) | | | |
| UDLS-L | 161.07 ± 75.69 | 239.38 ± 46.82 | 63.00/327.00 |
| ULS | 127.86 ± 14.81 | 205.33 ± 16.34 | 58.00/310.00 |
| UDLSA | 160.80 ± 69.36 | – | – |
| UDLS-C | 166.67 ± 27.54 | – | – |
| ULIFD | 392.50 ± 54.43 | – | – |
| CRP changes (mg/L) | | | |
| UDLS-L | 13.19 ± 4.70 | 20.26 ± 4.37 | 1.00/25.30 |
| ULS | 11.67 ± 2.56 | 18.87 ± 3.24 | 3.86/25.28 |
| UDLSA | 14.06 ± 2.45 | – | – |
| UDLS-C | 14.67 ± 4.04 | – | – |
| ULIFD | 46.19 ± 12.03 | – | – |

modified MacNab Score was 91.18% for UDLS-L, 92.85% for ULS, 90.00% for UDLSA, 66.67% for UDLS-C, and 90.00% for UDLIF.

# DISCUSSION

In this study, we explored the clinical efficacy and surgical invasiveness of various UBE-assisted surgical approaches in a large cohort of SDD patients who were subjected to follow-up observations, and for the first time presented the clinical outcomes of UBESS in patients with multilevel SDD. In addition, we also provided the experience of revision surgery after the failure of ULIFD (Fig. 4).

As a form of MISS, UBESS offers excellent performance with high operative flexibility and versatility (*Park et al., 2017*; *Gao et al., 2022*; *Kim et al., 2018*; *Kim et al., 2019*). Consistent with these reports, we have used UBE-assisted surgical approaches of UDLS-L, ULS, UDLSA, UDLS-C and UDLIF for the treatment of LDD, CSR and LIS in our center, and our experience and practice have demonstrated several advantages of UBESS.

Firstly, it offers rapid pain control and functional recovery. It was found in this study that postoperative ODI, back and leg VAS and JOA all improved significantly at each designated

Sun et al. (2025), *PeerJ*, DOI 10.7717/peerj.19076

**Table 7 Perioperative blood loss.**

| | Estimated total blood loss (ml) | | | Intraoperative blood loss (ml) | | | Postoperative blood loss (ml) | | | Hidden blood loss (ml) | | |
|---|---|---|---|---|---|---|---|---|---|---|---|---|
| | Single seg | Double seg | Triple seg | Single seg | Double seg | Triple seg | Single seg | Double seg | Triple seg | Single seg | Double seg | Triple seg |
| UDLS-L | 333.70 ± 180.01 | 575.69 ± 170.81 | 603.11 | 38.75 ± 30.70 | 81.25 ± 16.28 | 50.00 | 37.72 ± 34.14 | 45.94 ± 22.30 | 90.00 | 257.23 ± 163.78 | 448.50 ± 162.55 | 463.11 |
| ULS | 273.98 ± 42.59 | 434.28 ± 33.51 | 427.60 | 28.57 ± 20.35 | 68.33 ± 9.83 | 70.00 | 23.57 ± 8.52 | 36.67 ± 12.11 | 40.00 | 221.84 ± 56.43 | 329.28 ± 41.98 | 317.60 |
| UDLSA | 359.10 ± 68.67 | – | – | 37.50 ± 11.36 | – | – | 39.50 ± 10.66 | – | – | 282.10 ± 77.14 | – | – |
| UDLS-C | 255.96 ± 36.54 | – | – | 31.67 ± 5.77 | – | – | 20.00 ± 5.00 | – | – | 197.63 ± 42.30 | – | – |
| ULIFD | 672.56 ± 136.90 | – | – | 113.00 ± 24.52 | – | – | 86.00 ± 31.69 | – | – | 473.56 ± 108.35 | – | – |

**Table 8  Clinical outcomes of UBESS.**

| | UBESS (operative segment) | | | | |
|---|---|---|---|---|---|
| | UDLS-L (Single/Double/Triple) | ULS (Single/Double/Triple) | UDLSA (Single) | UDLS-C (Single) | ULIFD (Single) |
| Preoperative ODI | 50.78 ± 12.63/60.25 ± 9.50/63.00 | 51.43 ± 6.27/59.83 ± 3.66/77.00 | 51.80 ± 10.93 | – | 56.40 ± 6.88 |
| Postoperative ODI (1 month) | 29.29 ± 10.00/29.38 ± 4.01/47.00 | 29.57 ± 2.07/ 30.17 ± 4.44/57.00 | 30.70 ± 8.37 | – | 35.20 ± 9.22 |
| Postoperative ODI (3 months) | 17.51 ± 6.94/18.75 ± 3.06/28.00 | 17.88 ± 2.79/19.17 ± 1.17/38.00 | 16.30 ± 7.51 | – | 24.40 ± 10.76 |
| Postoperative ODI (1 year) | 9.43 ± 4.47/10.25 ± 3.94/12.00 | 9.71 ± 2.98/10.83 ± 1.47/20.00 | 8.30 ± 3.59 | – | 11.60 ± 3.40 |
| Preoperative JOA | – | – | – | 15/14/14 | – |
| Postoperative JOA (1 year) | – | – | – | 17/17/16 | – |
| Improvement rate (1 year, %) | – | – | – | 100/100/66.67 | – |
| Preoperative VAS back | 7.08 ± 0.89/7.88 ± 0.72/7.00 | 7.14 ± 1.07/7.67 ± 0.82/8.00 | 6.90 ± 1.20 | – | 6.10 ± 0.88 |
| Postoperative VAS back (1 month) | 1.98 ± 1.05/2.56 ± 0.73/5.00 | 1.86 ± 0.90/2.16 ± 0.75/5.00 | 1.90 ± 0.57 | – | 2.60 ± 1.17 |
| Postoperative VAS back (3 months) | 0.98 ± 0.58/1.25 ± 0.45/3.00 | 1.14 ± 0.69/1.33 ± 0.52/3.00 | 1.20 ± 0.42 | – | 1.60 ± 0.52 |
| Postoperative VAS back (1 year) | 0.82 ± 0.43/0.62 ± 0.50/1.00 | 0.71 ± 0.49/0.67 ± 0.52/2.00 | 0.60 ± 0.52 | – | 0.70 ± 0.48 |
| Preoperative VAS leg | 6.04 ± 0.94/6.94 ± 0.77/4.00 | 6.28 ± 0.76/6.83 ± 0.75/5.00 | 5.90 ± 0.99 | – | 6.30 ± 1.77 |
| Postoperative VAS leg (1 month) | 1.96 ± 0.87/2.31 ± 0.95/2.00 | 2.00 ± 0.82/2.33 ± 0.52/3.00 | 2.00 ± 0.47 | – | 1.90 ± 0.88 |
| Postoperative VAS leg (3 months) | 0.92 ± 0.52/1.31 ± 0.60/1.00 | 0.71 ± 0.49/1.17 ± 0.41/1.00 | 0.90 ± 0.32 | – | 1.20 ± 0.42 |
| Postoperative VAS leg (1 year) | 0.70 ± 0.50/0.94 ± 0.57/1.00 | 0.57 ± 0.53/0.67 ± 0.52/1.00 | 0.60 ± 0.52 | – | 0.50 ± 0.53 |
| Fusion rate (grade I, II) | – | – | – | – | 100% |
| Grade I | – | – | – | – | 6 |
| Grade II | – | – | – | – | 4 |
| Follow-up duration (months) | 14.41 ± 2.55/13.69 ± 1.25/13.33 | 13.14 ± 0.90/14.33 ± 1.63/15.60 | 13.70 ± 1.64 | 16.67 ± 0.58 | 15.40 ± 3.24 |
| Modified MacNab score (excellent/good/fair/poor) | 49/13/6/0 | 10/3/1/0 | 8/1/1/0 | 1/1/1/0 | 5/4/1/0 |
| Excellent/good rate (%) | 89.70 | 92.85 | 90.00 | 66.67 | 90.00 |

time point as compared with the preoperative conditions, as evidenced by short-term ODI and VAS pain scores for single level LDD and long-term JOA for single level CSR after operation, which is consistent with some recent studies (*Jiang et al., 2022*; *Huang et al., 2023*). *Gao et al. (2022)* reported that ODI decreased from 54.18 ± 10.42 to 36.89 ± 9.13, and the VAS score for back pain decreased from 5.71 ± 1.62 to 2.2 ± 30.68 in one week after ULIFD. In contrast, the clinical improvements in one month would be better in consideration of some factors such as ischemia reperfusion injury and complications. The clinical outcome of multilevel SDD with different operative approaches was comparable to that of single-level LDD. The similar improvement was also observed between single- and double- level diseases in patients using the same surgical approach. This may be attributed to multiple factors. Integrity of the spinal structure by directly reaching the target lesions, and safe and complete decompression under amplified visualization may be factors contributing to the rapid recovery. In addition, with minimal dissection and retraction of the muscles, muscle trophy and denervation could be avoided, which reduced the postoperative early pain (*Heemskerk et al., 2021*).

Secondly, it offers minimal invasiveness with considerable HBL. CK and CRP are commonly assumed to reflect muscle damage (*Thelander & Larsson, 1992*; *Kumbhare, Parkinson & Dunlop, 2008*). In our study, the CK level in multilevel LDD group was higher than that in the single-level LDD group. We found a dose–response relationship between CK and surgical invasiveness, which is consistent with the previous study (*Arts et al., 2007*). The ULIFD approach with more incisions and pedicle screw implantation in the confined working space may cause greater damage. Additionally, the pressure exerted on the paraspinal muscle may increase the CK level (*Kumbhare, Parkinson & Dunlop, 2008*). In the absence of infection, the trend of CRP change was found similar to CK, suggesting that the tissue damage was minimal, which is also evidenced by the lower level of both CK and CRP in LDD group with UL, because shorter OT of ULS and less surgical manipulation on the paraspinal structure in each group lowered the muscular exposure and trauma.

In spinal surgery, HBL is by no means negligible, because it constitutes a large percentage of TBL cases in this series (*Jiang et al., 2022*; *Peng et al., 2023*). It was also reported that HBL was the most important contributor to TBL in minimally invasive transforaminal lumbar interbody fusion (*Zhang et al., 2017*).

Our finding that HBL accounted for over 70% of TBL in each group is consistent with their findings. We also found HBL differences between groups with different surgical approaches for the same level SDD. The OT of different surgical approaches is classified as the risk factor of HBL, coupled with paraspinal muscle thickness (*Guo et al., 2022*). On the other hand, the HBL of ULIFD varies significantly, being 473.56 ± 108.35 ml. The variation of HBL of ULIFD ranged from 227.86 ± 221.75 ml to 472.19 ± 64.44 ml in previous studies (*Huang et al., 2023*; *Peng et al., 2023*; *Guo et al., 2022*). Our experience showed that HBL was much higher in the early learning period. But with the number of cases increasing, HBL will gradually decrease. *Kim et al. (2020b)* proposed that the caseloads for mastering this approach might be 34. However, *Xu et al. (2022)* advised 54 cases for proficiency. Therefore, the argument over the learning curve remains undefined.

Thirdly, it offers a low incidence of complication. Although complications occurred throughout hospitalization, it seemed acceptable clinically. All the complications occurred in the mastery phase. *Choi et al. (2016)* reported that the complication rate in the early learning period was about 30.8%. In the present study, the complication rate with UBESS for LDD was 3.26%, which is lower than the mean 6.7% (0%-13.8%) reported in a systemic review (*Lin et al., 2019*). Furthermore, symptomatic treatment is effective and associated with minimal damage to the surrounding tissues due to the high-resolution endoscopic technique. The complication rate of ULIFD for LIS was higher than that of UDLS-L, UDLSA and ULS for LDD in the early learning period, but there were only 10 LIS cases in our series. The more casse there are, the richer the content of the learning curve, which would help shorten the learning time for new learners who have never contacted this technique, which can be reflected by the steepness of the learning curve (*Xu et al., 2022*; *Ahn et al., 2022*). Besides, quality and safety of medical care should also be emphasized for the sake of achieving a stable success rate.

Dural tearing is the most common complication, as evidenced by the occurrence in previous studies (*Lin et al., 2019*). In our series, dural tearing occurred in two cases during the meningo-veteral ligament dissection in the early learning period. The absorbable gelatin sponge was applied on the dural breaks intraoperatively and conservative treatment was given postoperatively. Both patients spontaneously restored to health without headache after removing the drainage tube. We consider that dural sac adhesion to the spinal canal is the main reason of causing damage by radiofrequency or instruments. Furthermore, operational proficiency in the early learning period, disc fragments and loosened dura should also be considered (*Pan et al., 2020*). The debate on the appropriate treatment for dural tearing is still ongoing. Some studies recommend absolute bed stay and simple observation for minor tears of less than four mm, while open repair might be safe and effective for defects larger than 10 mm (*Park et al., 2020*; *Kim et al., 2020a*). But the repair technique under endoscopy requires further development. Thus, the detailed treatment regime for dural tearing by UBESS remains to be improved.

Several drawbacks of this study should be acknowledged. Firstly, the primary limitation of this study is the absence of a comparison group, which significantly affects the ability to draw definitive conclusions about the efficacy and safety of the operational technique being evaluated. Without a control group, it is challenging to determine whether the observed outcomes are genuinely attributable to the technique itself or if they could be influenced by other factors. Consequently, to provide a more robust assessment and validate the findings, prospective randomized controlled trials are necessary. Such studies would allow for a direct comparison, enhancing the reliability of the results and supporting evidence-based practice.

Secondly, a notable limitation of this study stems from its dependence on data sourced from a single center, which ultimately constrains the generalizability of its findings to a wider population. The lack of multi-center data support introduces potential biases that could arise from specific institutional practices, variations in patient demographics, and localized treatment methodologies. Consequently, the conclusions derived from this research may not adequately reflect the experiences or outcomes encountered in other

clinical environments. To bolster the evidence and improve the reliability of the results, future investigations should focus on incorporating multi-center data with larger sample sizes. This approach would yield a more comprehensive understanding of the operational technique and its effectiveness across a diverse spectrum of patient populations.

Thirdly, the relatively brief follow-up period in this study constitutes a significant limitation, as it may impede the comprehensive assessment of postoperative patients' functional recovery, along with the identification of complications and long-term prognosis. Insufficient follow-up duration can lead to a deficiency in critical data concerning the progression of patients over time, which is vital for accurately evaluating the effectiveness of the surgical intervention. Moreover, important late-onset complications may go undetected within such a constrained timeframe. To rectify this concern, future research should emphasize extending the follow-up duration, thereby facilitating a more in-depth exploration of the enduring outcomes and the overall impact of the treatment on patients' quality of life.

In addition, as this case series is dependent on retrospective observations, the inherent selection and measurement bias are unavoidable. Finally, a previous study reported that the lowest hematorit value appeared 3–5 days post-operation (*Yang et al., 2017*), and therefore estimation of blood loss by hematorit measured in the 2nd day after operation may be underestimated or overestimated.

## CONCLUSION

Our experience and practice demonstrate that treatment of spinal degenerative diseases with UBESS can offer excellent clinical, radiographic and functional outcomes. High operational flexibility and multi-scenario application with minimal invasiveness may contribute to the consideration of being the next generation of spinal endoscopic development following conventional endoscopic spinal surgery with one portal. It appears that UBESS is a rapid and curative treatment with a low incidence of complications for SDD. Nevertheless, HBL accounts for a large percentage of TBL in UBESS group. As controllable factors, lower HBL and TBL in UBESS group are conducive to the rapid recovery and shorter hospital stay.

**Abbreviation**

| | |
|---|---|
| **UBESS** | Unilateral biportal endoscopic spinal surgery |
| **BMI** | Body mass index |
| **Seg** | Segment/segments |
| **CSR** | Cervical spondylotic radiculopathy |
| **LDH** | Lumbar disc herniation |
| **RLDH** | Recurrent LDH |
| **IS** | Isthmic spondylolisthesis |
| **SS** | Spinal stenosis |
| **UDLS-L** | UBE-assisted lumbar discectomy+laminectomy+syndesmectomy |
| **UDLS-C** | UBE-assisted cervical discectomy+laminectomy+syndesmectomy |
| **UDLSA** | UBE-assisted lumbar discectomy+laminectomy+syndesmectomy+annulus fibrosus suture |

| ULS | UBE-assisted laminectomy+syndesmectomy |
| ULIFD | UBE-assisted lumbar interbody fusion+discectomy |

### Funding

The authors received no funding for this work. The APC was funded by a National Natural Science Foundation of China (NSFC) project titled "UGP-2 Integrin Glycosylation Signaling Axis-Mediated Mechanism of Malignant Behavior in Chondrosarcoma" led by Professor Haifeng Wei. The funders had no role in study design, data collection and analysis, decision to publish, or preparation of the manuscript.

### Grant Disclosures

The following grant information was disclosed by the authors:
National Natural Science Foundation of China (NSFC).

### Competing Interests

The authors declare there are no competing interests.

### Author Contributions

- Haitao Sun conceived and designed the experiments, performed the experiments, analyzed the data, prepared figures and/or tables, authored or reviewed drafts of the article, and approved the final draft.
- Qi Zhang conceived and designed the experiments, performed the experiments, analyzed the data, prepared figures and/or tables, authored or reviewed drafts of the article, and approved the final draft.
- Kehan Xu conceived and designed the experiments, performed the experiments, authored or reviewed drafts of the article, and approved the final draft.
- Zihuan Zhou performed the experiments, authored or reviewed drafts of the article, and approved the final draft.
- Dongjie Jiang performed the experiments, authored or reviewed drafts of the article, and approved the final draft.
- Shaohui He performed the experiments, authored or reviewed drafts of the article, and approved the final draft.
- Jilu Liu conceived and designed the experiments, performed the experiments, authored or reviewed drafts of the article, and approved the final draft.
- Haifeng Wei conceived and designed the experiments, performed the experiments, authored or reviewed drafts of the article, and approved the final draft.

### Human Ethics

The following information was supplied relating to ethical approvals (i.e., approving body and any reference numbers):
    The Naval Hospital of Eastern Theater ethics committee.

## Data Availability

The raw data is available in the Supplementary File.

## Supplemental Information

Supplemental information for this article can be found online at http://dx.doi.org/10.7717/peerj.19076#supplemental-information.

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
