# Peer review of "A retrospective study on safety and clinical outcomes of unilateral biportal endoscopic technique for spinal degenerative diseases"

_PeerJ, doi:10.7717/peerj.19076_

## Round 0.1 · original submission · Major Revisions

Our apologies for the delayed process. We have now received 3 review reports and there is consistent feedback that the experimental design (i.e. including disparate surgical techniques) needs to be significantly re-visited in order to justify any conclusions. We suggest you consider (for example) limiting the inclusion criteria to lumbar decompressions only; or (as Reviewer 1 indicated) to consider comparing one, two, and multilevel endoscopic lumbar decompressions.

Reviewer 1 ·

Basic reporting

Abstract, results section, acronym ULIFD is not defined - Page 1 Line 22
"surgical contradictions" should be "contraindications" - Page 3 Line 70
"favum" should be "flavum" - Page 4 line 94

Experimental design

This is a retrospective, cross-sectional study to present data on the invasiveness of various biportal endoscopic techniques for the treatment of cervical and lumbar degenerative disorders. Inter-group comparisons of one and multilevel procedures were performed.

Validity of the findings

The purpose of this study is well stated, however the validity is questionable. The study group is incredibly heterogenous; it is not appropriate to group cervical discectomy with lumbar fusion patients when evaluating invasiveness. Likewise, there is a high variety of surgical techniques employed in the lumbar cohort as well ranging from laminotomy with discectomy and/or annular repair, to interbody fusion with instrumentation. The addition of these techniques limits the applicability and validity of this study. A more appropriate presentation would be decompressive surgeries in the lumbar spine alone comparing one and two or multilevel procedures.

Reviewer 2 ·

Basic reporting

Title: A long-term retrospective study on safety and clinical outcomes of unilateral biportal endoscopic technique for spinal degenerative diseases

Outline: This retrospective study aims to summarize the clinical outcome and surgical invasiveness of UBE for the treatment of various spinal degenerative diseases in a single center to validate the safety and application value of UBESS. In this study, the authors concluded that UBESS has proved to be a safe, reliable and minimally invasive option for spinal degenerative diseases, with significant benefits of pain control, rapid functional recovery, short hospitalization, and early rehabilitation. However, postoperative hidden blood loss should be put under the careful management when performing UBESS.

I have no comment for basic reporting. However, several concerns were presented.

Experimental design

1. The authors emphasized that this study is long-term follow-up. However, in this study, there was no definition of long-term follow-up. Also, only one-year follow-up was presented in this study. The results of UBESS are common on the literature.

2. In this study, postoperative complications (DVT ...) were not included.

3. This study is not comparative study, which can't justify the conclusion of this study.

Validity of the findings

I think current version is relatively weak, which need to be supplemented on comparative (open procedures) and long-term data (more than 5-year follow-up)

Additional comments

No additional comments

·

Basic reporting

The authors have satisfactorily described and outlined their findings with pre-defined outcome measures using standard measures such as operation time, hospital stay, blood loss and postoperative immobilization time. The language is serviceable, references are sighted and the article has an acceptable structure.

Experimental design

Measurement of outcomes using the visual analogue score is reasonable. However, utilizing the mean clinically important difference as defined the literature would provide readers with more confidence in the authors’ finding. The surgical endoscopic technique is well described.

Validity of the findings

An original study which continues to validate endoscopic spinal surgery as a viable and even superior option in certain cases.

Additional comments

The authors have performed a well conducted study and are to be congratulated. A few considerations.
1. A tabulated and written summary of similar studies in this area and their findings on functional outcome, hospitalization stay and blood loss should be compared.
2. Do the authors that only those operations in whom a straightforward endoscopic access were included which introduced bias? For example more complex cases or those not amenable to endoscopic access were therefore selected for traditional open surgery.
3. The authors need to discuss the limitations of their study and their own learning curve in greater detail.

---

## Round 0.2 · accepted · Accept

I am writing to inform you that your manuscript - A retrospective study on safety and clinical outcomes of unilateral biportal endoscopic technique for spinal degenerative diseases - has been Accepted for publication. Congratulations!

Reviewer 1 ·

Basic reporting

Meets criteria

Experimental design

Retrospective cross sectional design aiming to delineate invasiveness, safety, and effectiveness of UBESS in various forms. No comparison groups.

Validity of the findings

Validity is acceptable.

Additional comments

This manuscript presents the authors' experience with UBESS. A variety of pathologies and surgical approaches are presented together with the purported goal of defining the invasiveness, safety, and efficacy of UBESS. It is not clear to me how these novel evaluations of blood loss, which is minuscule, are a useful way of defining invasiveness. WIth no comparator group, it is not obvious how much less invasive UBESS is compared to, for example, a tubular TLIF for degenerative spondylolisthesis.

Taken as a narrative review of the authors' experience with UBESS, I feel that this does achieve its stated aim of demonstrating that the procedure is safe and effective, however without comparison groups to other minimally invasive techniques with a narrow focus, the impact of this paper is questionable.

·

Basic reporting

Language, referring and article structures is satisfactory.

Experimental design

Research design and investigation is reasonable.

Validity of the findings

The findings are well supported by the methodology